# Cultural Adaptation of a Health Literacy Toolkit for Healthcare Professionals Working in the Primary Care Setting with Older Adults

**DOI:** 10.3390/healthcare11050776

**Published:** 2023-03-06

**Authors:** Areti Efthymiou, Argyroula Kalaitzaki, Michael Rovithis

**Affiliations:** 1Department of Social Work, Hellenic Mediterranean University (HMU), 71410 Heraklion, Crete, Greece; 2Laboratory of Interdisciplinary Approaches to the Enhancement of Quality of Life (Quality of Life Lab), Hellenic Mediterranean University (HMU), 71410 Heraklion, Crete, Greece; 3University Centre of Research and Innovation ‘Institute of AgriFood and Life Sciences, Hellenic Mediterranean University (HMU), 71410 Heraklion, Crete, Greece; 4Department of Business Administration and Tourism, Hellenic Mediterranean University (HMU), 71410 Heraklion, Crete, Greece

**Keywords:** communication, health literacy, healthcare professionals, older adults, self-efficacy, toolkit

## Abstract

Healthcare professionals’ health literacy (HL) knowledge and skills influence their interaction with older adults. Healthcare professionals, when effectively communicating with older adults, can empower and enhance patients’ skills to make informed decisions about their health. The study aimed to adapt and pilot test a HL toolkit to enhance the HL skills of health professionals working with older adults. A mixed methodology of three phases was used. Initially, the healthcare professionals’ and older adults’ needs were identified. Following a literature review of existing tools, a HL toolkit was selected, translated, and adapted into Greek. The HL toolkit was introduced to 128 healthcare professionals as part of 4 h webinars; 82 healthcare professionals completed baseline and post assessments, and 24 healthcare professionals implemented it in their clinical practice. The questionnaires used included an interview on HL knowledge, communication strategies, and self-efficacy using a communication scale. HL and communication strategies knowledge (13 items) and self-efficacy in communication (t = −11.127, df = 81, *p* < 0.001) improved after the end of the HL webinars, and improvement was retained during the follow-up after 2 months (H = 8.99, df = 2, *p* < 0.05). A culturally adapted HL toolkit was developed to support the needs of healthcare professionals working with older adults, taking into consideration their feedback in all phases of the development.

## 1. Introduction

According to WHO, strengthening the health literacy (HL) skills of the general population is a multicomponent process involving the participation of healthcare organizations, healthcare professionals, healthcare users, and the environment [1]. An improvement in healthcare organizations’ and healthcare professionals’ HL skills could be beneficial for national healthcare systems, considering that studies have associated patients’ low HL with higher medical costs [2,3].

Older adults are considered a vulnerable population with low HL [4], who may face cognitive, sensory, and other physical challenges that prevent them from communicating effectively with healthcare professionals [5]. Older adults’ low HL is associated with negative health outcomes, such as deteriorated physical and cognitive functioning, worse mental health, low medication adherence, worse disease management, fewer health-related preventive actions, longer hospital stays, frequent emergency admissions, and overall higher mortality rates [3,4].

Among the objectives of “Healthy People” for 2030, an initiative running for almost thirty years by the US Department of Health and Human Services, are access to healthcare services and an improvement in communication among healthcare providers and patients (i.e., involve patients in decision-making, assure understanding, increase clear communication) [6]. Older adults are coping with barriers in accessing and utilizing healthcare services [7]. Factors influencing access to healthcare services in primary care include individual (literacy and education, health beliefs, limited mobility, limited digital skills), socioeconomic, cultural/linguistic, environmental (geographical location, transportation, visit modality), and organizational (staff shortage) factors [8,9,10]. Healthcare professionals play a crucial role in assisting healthcare users to overcome the aforementioned barriers, as it was stated in Ottawa Chart [11], through the redistribution of power, participation, and healthcare users becoming experts.

The knowledge and comprehension of HL among healthcare professionals influence the communication between healthcare professionals and older individuals with chronic diseases and their families [5]. Communication and empowerment of the high-risk population are considered one of the ten attributes of HL organizations [12]. Organizational HL is gaining prominence within health organizations. The ten attributes of HL organizations include [12] leadership to promote HL; integration of HL in planning; evaluation measures; training of the workforce on HL issues; participation of healthcare users in the design, implementation, and evaluation of the health services; combating stigma; use of HL strategies in communication; accessible services and health resources provided by the organization; clear communication; and empowered users in high-risk situations (medication adherence) [12].

The use of clear language without medical jargon, the use of a person-centered approach, and the implementation of communication strategies according to healthcare professionals working with older adults improved communication with them [13]. According to the HL Universal Precautions, healthcare professionals should not assume the HL level of their patients based on their appearance [14]. The Agency of Healthcare and Research Quality developed a universal precautions toolkit for physicians including 20 tools [15], which was later updated to 21 tools [16]. The main categories included were raising awareness, improving spoken and written communication, and improving supportive systems [16]. To the authors’ knowledge, there is no HL toolkit adapted to the needs of healthcare professionals working with older adults [17]. The study aimed to adapt and pilot a health literacy toolkit to enhance the HL skills of healthcare professionals working with older people to validly detect and empower patients’ HL levels.

## 2. Materials and Methods

### 2.1. Study Design

The current study followed a mixed method design with three phases lasting 24 months (November 2020 to October 2022) and following the cultural adaptation process model (CAP): (1) examining the target groups’ needs, (2) developing the adaptation methodology, and (3) adapting the selected tools and pilot testing them [18]. Target groups of the adaptation process were healthcare professionals working with older adults (Figure 1).

#### 2.1.1. Examining the Target Groups’ Needs

*Identify related literature*. A scoping review was conducted by the research group for the period 2000 to 2020 to identify tools for translation and adaptation, HL training courses, and evaluation measures. The literature search of five electronic databases (PubMed, MEDLINE, CINAHL, PsychInfo, and Opengrey) resulted in 27 papers, and it was concluded that there is a lack of HL tools tailored for healthcare professionals working with older healthcare users [17]. The most used HL toolkit was the HLUPT. Information is available in the published scoping review [17].

*Identify healthcare professionals’ and older healthcare users’ needs.* Two focus groups were conducted to assess the HL knowledge and communication needs of healthcare professionals (*n* = 7) and older healthcare users (*n* = 5). Questions included topics on HL knowledge, perspectives on barriers and facilitators in patient–healthcare professional interaction, and healthcare professionals’ training needs [13]. Healthcare professionals reported that trust, collaboration, patients’ education, psychological resilience, carers’ participation, use of plain language, and compassion facilitated their interaction with older healthcare users. On the other hand, the rigid lifestyle of older adults and any cognitive, emotional, sensory, and physical health issues comprised barriers. Older healthcare users considered as important elements in their interaction with healthcare professionals the appropriate preparation before their visit, the assessment of the severity of their health problems, their carers’ participation, nonuse of medical jargon, patients’ involvement in decision-making, equality, respect, compassion, and healthcare professionals’ encouraging discussion [13].

#### 2.1.2. Developing the HLUPT Adaptation Methodology

Permission to adapt the health literacy toolkit was granted by Advancing Excellence in Health Care. A certified translator from English to Greek and a group of HL experts were identified to evaluate the terminology, and cultural appropriacy of the selected tools, supplementary material, and inclusion criteria for the tools were agreed upon. The inclusion criterion set was that the tool to be in accordance with national health system processes in Greece.

This pilot test method has already been used in the adaptation of HLUPT for cardiology and rheumatology [19]. Healthcare professionals were initially informed about the HLUPT tools during a 4 h webinar and were then asked to use the toolkit for a two-month period. The webinars were delivered online for two main reasons: to facilitate participation from different regions of Greece and to ensure safety due to pandemic restrictions. Overall, six webinars of 12–20 participants each were delivered from October 2021 to June 2022.

The content of the webinars derived from the work by Kripalani et al. [14] with permission to use and adapt the workshop material, “Strategies to improve communication between pharmacy staff and patients: A training program for pharmacy staff curriculum guide”. The webinar included a theoretical section in the form of a presentation on the definition of HL, red flags for identifying older patients with low HL, and an introduction to the HL toolkit. The second section outlined the most important communication strategies (plain language, teach-back, summarizing the main message, and searching for online information). The third section consisted of role-playing exercises. Evaluation of the knowledge and communication skills was designed at baseline, after the end of the webinar, and for the small sample that implemented the adapted HLUPT after the end of a two-month period.

#### 2.1.3. HLUPT Adaptation

In collaboration with the research team, a group of four HL experts evaluated the terminology and cultural appropriacy of the selected tools and supplementary material. The research team tailored the language to be culturally sensitive to Greek content and values and ensured the content was applicable to healthcare professionals working with older adults according to the ecological validity model (16).

In the pre-piloting phase, the research team excluded information regarding private insurance forms, nonmedical support and medical resources, referrals, and literacy and math resources (Table 1).

### 2.2. Sample and Recruitment Process of the HLUPT Pilot Phase

Over 250 professionals were invited to participate in the pilot phase of the HLUPT. Recruitment followed a convenience and snowball sampling procedure, according to which professionals invited colleagues via social media, online groups, healthcare services, and university departments. The eligibility criteria for the healthcare professionals included: being a physician, nurse, psychologist, or other healthcare professional, speaking and writing in Greek, and working in a primary healthcare setting (e.g., hospital, day center, day clinic). Initially, 128 healthcare professionals registered to attend the webinar, and 82 completed the post-training assessment. The most common explanations for drop-out were time constraints and an abundance of responsibilities. Other reasons concerned organization barriers and the inability to comprehend the task of this phase. After the webinar, a small sample of healthcare professionals (*n* = 24) implemented the adapted HLUPT in their clinical practice for a two-month period. The healthcare professionals used up to three tools in their daily clinical practice for two months under the supervision of the research team. Short calls were scheduled with the researchers for questions or clarification. At the end of this two-month period, healthcare professionals evaluated their communication skills and knowledge of HL and participated in three focus groups.

Three focus groups were organized with 15 participants. The discussion was facilitated with the use of a guide that included questions regarding the HL toolkit and the way they used the selected HL tools (i.e., how they implemented the tools, barriers to implementation, evaluation of the practicality of the tools, suggestions of HL tools from the toolkit, written language of the toolkit, organization and staff attitudes toward the toolkit, and future training needs). The focus group sessions lasted between 40 and 60 min. The principal author (A.E.) served as a moderator and the second author (A.K.) as an observer.

### 2.3. Questionnaires Used in the HLUPT Pilot Phase

Demographic information (e.g., gender, age, education, profession, professional experience, and type of healthcare service) was obtained from healthcare professionals at baseline. To assess HL knowledge and communication skills, 13 Likert-type questions were adapted based on the research conducted by Mackert [20], including HL perceived knowledge, perceived ability to identify low-HL patients, communication skills, and one open question on the definition of HL.

Self-Efficacy-12 questionnaire was validated in Greek for the purpose of pilot test phase [21]. The questionnaire consists of 12 items measuring healthcare professionals’ self-efficacy of skills used during the patient–clinician encounter, rated on a 10-point Likert scale ranging from 1 = very uncertain to 10 = very certain. The sum of scores ranges from 12–120, and higher scores indicate higher levels of self-efficacy in skills [22]. The first 10 items measure “self-efficacy in communication skills and strategies”, and the last two items measure “self-efficacy of successful interaction”. Both factors of the Greek version have shown high internal consistency (factor 1: α = 0.95 and factor 2: α = 0.93), similar to the total scale (α = 0.96).

### 2.4. Statistical Analysis

Descriptive statistics and dependent sample t-tests were computed to compare the pre- and post-assessments. The Kruskal–Wallis test was used to compare baseline, post-, and follow-up scores for the sample of 24 healthcare professionals. Three focus group discussions evaluated the usefulness of the HL toolkit after implementing the selected tools for a two-month period. Transcribed scripts, derived from the focus group discussions, were content analyzed.

### 2.5. Ethics and Informed Consent

Permission to conduct the study was granted by the bioethics committee of the Hellenic Mediterranean University (63/EMΠ 95). All participants were fully informed of the study aims and the requirements for their participation in the study. Consent forms were signed before the pilot testing, and all participants were informed that they could withdraw their participation and that their data could be excluded by contacting the main investigator. Researchers in all phases of the study promoted the wellbeing of the participants. The researcher/trainer tried to make participants feel comfortable and resolve any kind of conflict. To safeguard sensitive personal data, a database protected by password was developed and stored on the research team’s university computers; only members of the research team had access to the database.

## 3. Results

### 3.1. Descriptive Characteristics

Most of the participants who attended the webinars were women (*n* = 120, 93.8%) with tertiary and postgraduate education (N = 105, 82%), a mean age of 44 years old, and worked in public healthcare services (*n* = 103, 83%). Nurses (*n* = 36, 28.1%), health visitors (*n* = 30, 23.4%), and social workers (*n* = 22, 17.2%) were the professions with the highest representation in the sample. Physicians, physiotherapists, and occupational therapists had the lowest participation. Almost half of the sample had over 10 years of experience with older healthcare users (Table 2).

### 3.2. HL Definition, Knowledge, and Communication Self-Efficacy

More than two-thirds of the healthcare professionals (*n* = 83, 64%) provided at baseline the HL definitions, whereas the remaining gave vague definitions of health, skills, knowledge, clinical diagnosis, and communication skills of healthcare professionals. Most of the respondents identified the ability to comprehend medical recommendations and treatment plans in order to make health decisions as the central aspect of the HL definition, while others identified the ability to comprehend medical jargon. Examples of the HL definitions follow:

“the skill to understand medical jargon and to use the internet to support your health”;

“the skill to understand the therapeutic plan”;

“the skill to understand and process health information to make health decisions”.

In the post-test assessment (after the end of the webinar), 82 participants had statistically significantly improved their HL knowledge in all 13 of Mackert’s interview items (Table 3).

There was a significant difference in SE-12-Gr between the baseline scores (M = 92.1, SD = 13.62) and the post-assessment scores for the 82 participants (M = 104.24, SD = 10.82) (t = −11.127, df = 81, *p* < 0.001) (Table 4).

### 3.3. Implementation of HLUPT Tools in Clinical Practice

The sample that continued to the implementation of the three tools from the HLUPT for healthcare professionals working with older adults retained a higher rank mean (39.17) in the SE-12-Gr at the follow-up assessment compared with baseline (26.42) (Table 5).

### 3.4. Focus Group Discussions

Of the 15 participants in the three groups, 29% (*n* = 7) used the HL toolkit for less than a week, 25% (*n* = 6) for 1 to 2 weeks, and the remaining participants (46%; *n* = 11) from 1 to 2 months. Half of the participants in the focus groups reported using the HL toolkit frequently during the aforementioned periods.

Three themes were discussed: (1) the preferred tools, (2) ways to improve the toolkit, and (3) HL toolkit usefulness and health organization acceptance.

*Preferred tools*. The most preferred tool was “Tool 3. Teach-back method” (n = 9, 60%). Participants reported that they informed their coworkers, and they planned meetings to do so whenever possible using available material and instructions from “Tool 1. Raise awareness” of the HLUPT. Healthcare professionals working at home preferred “Tool 4. Brown bag medication review” and “Tool 10. Assisting older adults with their medical treatment.” “Tool 2. Clear communication” was also frequently selected and usually combined with “Tool 3”. Only in one case did a healthcare professional living and working in a rural area favor “Tool 6. Cultural and linguistic differences.”*Ways to improve the toolkit*. Participants in the focus group suggested ways to enhance the HL toolkit:
-to utilize exercises and images to facilitate comprehension of the tools;-to include components for older adults with memory deficiencies and dementia in “Tool 3. Teach-back method”;-to introduce the concept of healthy and active aging;-to modify the toolkit for use by people working in the public and private sectors (e.g., banks, supermarkets, post offices);-to incorporate suggestions for healthcare professionals working with older adults who are illiterate in “Tool 10. Assisting older adults with their medical treatment”;-to include a friendly service environment adaptation tool.*HL toolkit usefulness and health organization support*. All the participants in the focus group assessed the HL toolkit as very useful and its resources as comprehensible. They acknowledged that they had to acquire this knowledge through practice with older adults. Participants reported that the toolkit could be a valuable resource for young healthcare professionals starting their careers and for older people to gain new knowledge. They discussed the difficulty in establishing an HL team within their organization because of social distance measures and relevant restrictions imposed by the COVID-19 pandemic (i.e., could not meet regularly with their colleagues). They also had difficulty approaching their experienced colleagues. In addition, frequent changes in supervisors/managers in public healthcare services posed a potential barrier to their efforts to raise HL awareness among the organization’s staff. The final adapted toolkit is available in Table 6.

## 4. Discussion

The purpose of the study was the adaptation of a HL toolkit for healthcare professionals working with older adults. This study provided a toolkit written in plain language with examples of everyday clinical practice provided by healthcare professionals. After the 4 h webinars in the framework of the toolkit piloting, healthcare professionals improved their communication self-efficacy and HL knowledge, and this finding remained after the 2-month follow-up for the healthcare professionals who implemented the toolkit in their clinical practice.

There are HL training courses and tools targeting healthcare professionals and physicians [14,23,24,25,26], but there is a lack of courses and tools targeting healthcare professionals working with older adults [17]. Older adults are considered an at-risk population with low HL, facing difficulty in adhering to the therapeutic plan and effectively communicating with healthcare professionals [4,5,27]. They face multiple health issues and are usually excluded by healthcare professionals from the decision-making process [28,29].

The findings of this study were valid and reliable. The three stages of the cultural adaptation process model [23] were as follows: setting the scene, planning the adaptation, and proceeding with the adaptation and pilot test. Our findings are in line with Mackert et al.’s [20] findings, according to which, on the one hand, participants overestimated their HL knowledge, and on the other hand, their knowledge improved after a training course. Similar studies assessing self-efficacy in communication found that it was improved after communication skills training of physicians and nurses [29]. Two-thirds of the participants in the present study were aware of and identified aspects of the HL concept (e.g., understanding health information, making health decisions, communication with physicians, and medical adherence).

Healthcare professionals perceived the HL toolkit as a valuable, easy-to-use resource in everyday clinical practice, and they contributed their expertise to the focus groups to improve the toolkit. The most preferred tools included “Tool 1. Raise awareness,” “Tool 2. Communicate clearly,” “Tool 3. Teach-back method,” “Tool 5. Brown bag medication review,” and “Tool 10. Assisting older adults with their medical treatment.” Callahan et al. [19] recognized the teach-back method and brown bag medication review as the most frequently used tools to raise awareness among staff. Older adults’ medication adherence is considered an important dimension of the healthcare professionals’ clinical work and usually is reported as the most difficult issue to handle in the older adults’ caregiving process [30].

Limitations of this study were the small number of medical professionals in the adaptation process and the small number of participants in the follow-up phase. For this reason, the results of the study concerning the specific sample could not be generalized to other healthcare professionals. The strength of the study included the involvement of healthcare professionals in all phases of the adaptation process and providing a toolkit tailored to the healthcare professionals’ needs. During the focus group discussions with older healthcare users regarding their interaction with healthcare professionals, they mostly focused on physicians’ input [13], and physicians were the least represented healthcare professional group in our sample. In addition, lack of funding prevented the linguistic adaptation of audiovisual materials that were included in the HLUPT. All video links were added to the toolkit, along with a note instructing users to enable subtitles when applicable.

Primary healthcare in Greece is delivered through public and private services, and healthcare users need to find and appraise healthcare services without any support from the healthcare system [31]. They need to decide on the proper healthcare professional specialty and healthcare department [31]. HL is a concept not fully appraised by health organizations in Greece, even if more and more discussions on the topic have been raised after the COVID-19 pandemic.

Future research could focus on the effectiveness of the HL toolkit within the context of healthcare professional practice and the adaptation of the toolkit to other high-risk populations (e.g., migrants). HL training among healthcare professionals builds networks, promotes HL leadership, empowers professionals, and builds HL organizations [32].

## 5. Conclusions

Public healthcare services could adopt the HL toolkit to support the training needs of primary care healthcare professionals [32]. We consider that the HL toolkit provides a set of 11 tools tailored to the needs of healthcare professionals working in the Greek context. The HL toolkit could assist healthcare professionals in their everyday clinical work, be part of vocational and educational seminars, and be promoted within the healthcare organizational context. The use of the HL toolkit will facilitate the clinical work of healthcare professionals working with older adults, and indirectly, it is expected to enhance the HL of older adults.

## Figures and Tables

**Figure 1 healthcare-11-00776-f001:**
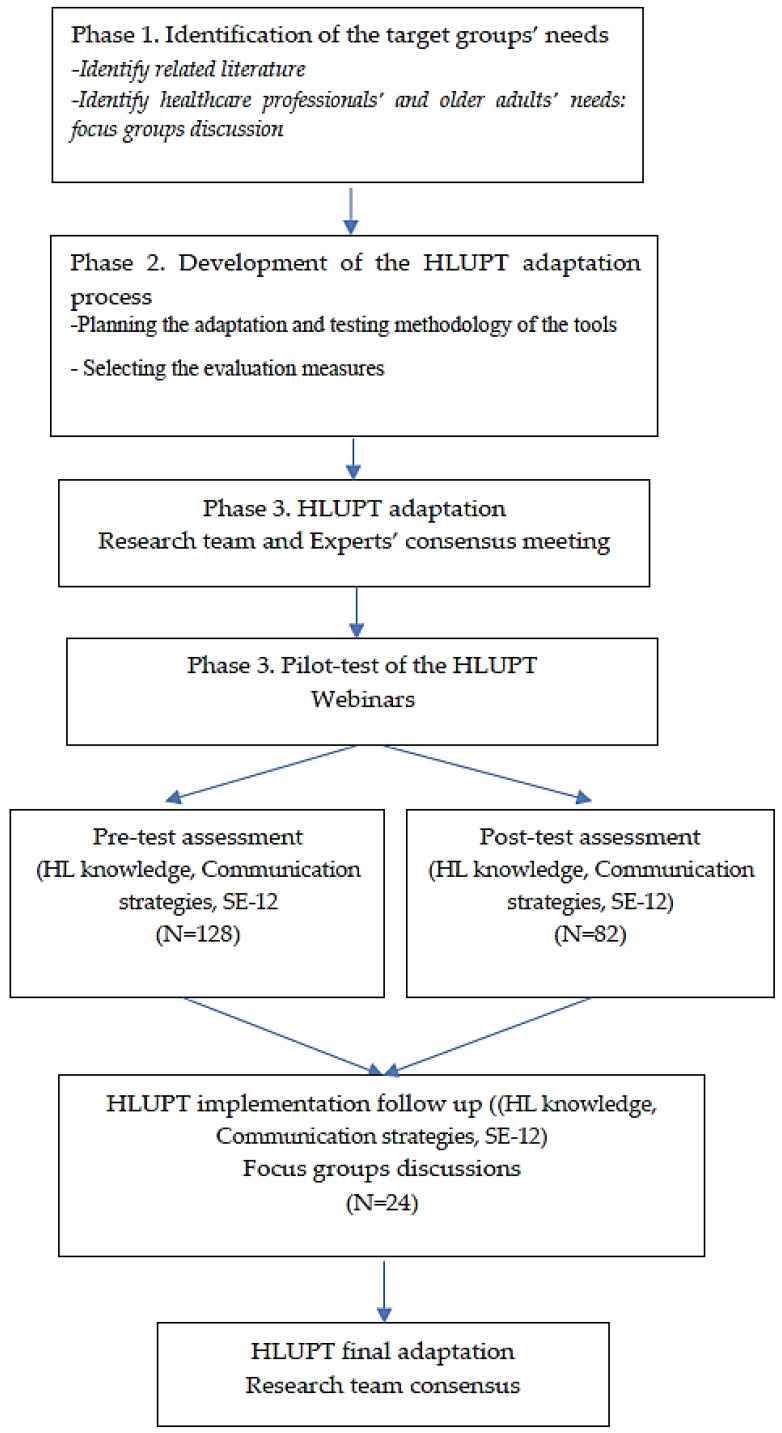
HLUPT cultural adaptation process.

**Table 1 healthcare-11-00776-t001:** Health Literacy Universal Precautions Toolkit (HLUPT) selected tools before piloting.

	Selected HLUPT to Be Adapted for Healthcare Professionals Working with Older Adults	Tools Selected for Piloting	Content Modifications Focusing on Healthcare Professionals Working with Older Adults
1	Tool 1. Form a team		Partially included in Tool 3
2	Tool 2. Create a health literacy improvement plan		Not included
3	Tool 3. Raise awareness (path of improvement)	Tool 1. Raise awareness (path of improvement) + annexes	Use of simple language focusing on healthcare professionals working with older adults
			Presentation adapted to Greek context
			Role-playing adapted to Greek context
			Links updated for Greek context, including training resources for healthcare professionals
			HL quiz adapted to Greek context
4	Tool 4. Communicate clearly (spoken communication)	Tool 2. Communicate clearly (spoken communication) + annexes	Use of simple language focusing on healthcare professionals working with older adults
			Person-centered communication strategies and tailored communication tips for older adults focusing on older adults were added
			Links to videos for person-centered care were added
			Adaptation of annexes in Greek
5	Tool 5. Teach-back (spoken communication)	Tool 3. Teach-back (spoken communication)	Use of simple language focusing on healthcare professionals working with older adults
			Links subtitled in Greek
			Links to teach-back training.org
			Modifying the person experience targeting older patients
			Evidence added in relation to the implementation of the technique for older adults
6	Tool 6. Follow-up with patients		Not included
7	Tool 7. Improve telephone access		Not included
8	Tool 8. Brown bag medicine reviews (spoken communication)	Tool 4. Brown bag medicine reviews (spoken communication)	Use of simple language focusing on healthcare professionals working with older adults
			Including tips for polypharmacy, older adults’ rigid attitudes in the use of medication, naming of medication. Exclusion of terms irrelevant to Greek context (medicine reconciliation), American Institute manual, Ohio toolkit
9	Tool 9. Address language differences		(Partially included in tool of cultural customs and beliefs)
10	Tool 10. Consider culture, customs, and beliefs, incl. points for language differences (spoken communication)	Tool 5. Consider culture, customs, and beliefs, incl. points for language differences (spoken communication)	Use of simple language focusing on healthcare professionals working with older adults
			Points from the tool of language differences were added
			Associations working with refugees and migrants and associations working with older adults in Greece were added
			Links to eLearning courses for cultural skills in Greek language
11	Tool 11. Assess, select, and create easy-to-understand materials (written communication)	Tool 6. Assess, select, and create easy-to-understand materials (written communication)	Use of simple language focusing on healthcare professionals working with older adults
			Use of HLS-EU evidence on health literacy and of OECD for literacy in Greece
			Link of a Greek readability online tool developed by the Centre of Greek language
			Links to eLearning courses and manuals to enhance digital skills of older adults (in Greek) and to use applications
12	Τοοl 12. Use health education material effectively (written communication)	Τοοl 7. Use health education material effectively (written communication)	Updating terminology focusing on Web 2.0
			Links for the eLearning course eLILY1 focusing on ehealth literacy skills of carers working with older adults
			Exclusion of nonrelevant examples
13	Tool 13. Welcome patients: helpful attitude, signs, and more		Not included
14	Tool 14. Encourage questions (self-management and empowerment)	Tool 8. Encourage questions (self-management and empowerment) + annexes	Use of simple language focusing on healthcare professionals working with older adults
			Type of questions that older adults should ask when visiting a clinical practice
15	Tool 15. Make action plans		Not included
16	Tool 16. Help patients remember how and when to take their medicine (self-management and empowerment)	Tool 9. Help patients remember how and when to take their medicine (self-management and empowerment) + annexes	Use of simple language focusing on healthcare professionals working with older adults
			Tips to help older people to take their medication (memory game, naming game)
			Links to medication reminder applications
17	Tool 17. Get patient feedback (self-management and empowerment)	Tool 10. Get patient feedback (self-management and empowerment) + annexes	Use of simple language focusing on healthcare professionals working with older adults
18	Link patient to nonmedical support		Not included
19	Direct patients to medicine resources		Not included
20	Connect patients with literacy and math resources		Not included
21	Make referrals easy		Not included

**Table 2 healthcare-11-00776-t002:** Descriptive characteristics of the participants.

	Variable	N (%)	N (%)	N (%)	N (%)
		Pretraining (*n* = 128)	Post-Training (*n* = 82)	Follow-Up (*n* = 24)	Focus Group (*n* = 15)
Sex	Women	120 (93.8)	77 (93.9)	23 (95.8)	14 (93.3)
	Men	8 (6.3)	5 (6.1)	1 (4.2)	1 (6.7)
Education	Secondary/upper secondary	17 (13.3)	9 (11)	3 (12.5)	1 (6.7)
	Tertiary	52 (40.6)	35 (42.7)	11 (45.8)	8 (53.3)
	Postgraduate	53 (41.4)	35 (42.7)	10 (41.7)	6 (40)
	Doctoral	6 (4.7)	3 (3.7)	0	0
Age	43.16	44.63 (SD = 7.07)	46.21	46	
	(SD = 8.76)		(SD = 7.25)	(SD = 7.25)	
	Physician	3 (2.3)	1 (1.2)	0	0
Profession	Nurse	36 (28.1)	25 (30.5)	5 (20.8)	4 (26.7)
	Social worker	22 (17.2)	15 (18.3)	5 (20.8)	5 (33.3)
	Physiotherapist	1 (.8)	0	0	0
	Nurse assistant	7 (5.5)	2 (2.4)	1 (4.2)	0
	Psychologist	14 (10.9)	11 (13.4)	2 (8.3)	2 (13.3)
	Health visitor	30 (23.4)	16 (19.5)	5 (20.8)	0
	Other (occupational therapist, linguist, sociologist)	15 (23.4)	12 (14.6)	6 (25)	4 (26.7)
Type of service	Private	21 (16.9)	13 (15.9)	4 (16.7)	2 (13.3)
	Public	103 (83.1)	69 (84.1)	20 (83.3)	13 (86.7)
Professional experience	<1 year	9 (7.1)	3 (3.7)	1 (4.2)	0
	1—less than 2 years	16 (12.5)	7 (8.5)	1 (4.2)	1 (6.7)
	2—less than 5 years	20 (15.6)	11 (13.4)	2 (8.3)	1 (6.7)
	5—less than 10 years	12 (9.4)	9 (11)	1 (4.2)	1 (6.7)
	>10	69 (53.9)	51 (62.2)	18 (75)	12 (80)

**Table 3 healthcare-11-00776-t003:** Mackert’s interview items (*n* = 82).

	Pre-Assessment		Post Assessment		
	M	SD	M	SD	*t*-Test (df)
Q1. I understand what it means for patients to have low health literacy	5.74	1.3	6.63	0.59	−6.532 *** (80)
Q2. I know the prevalence of low health literacy in Greece	4.24	1.65	6.01	1.04	−8.709 *** (80)
Q3. I know the groups that are more likely to be low health literate	5.8	1.15	6.43	0.72	−8.579 *** (80)
Q4. I understand the health outcomes associated with low health literacy	5.19	1.28	6.59	0.58	−6.726 *** (80)
Q5. Identifying low health literate patients	5.19	1.28	6.21	0.77	−6.963 *** (80)
Q6. Paying attention to whether or not my patients understand what I’m telling them	5.77	0.98	6.45	0.63	−6.457 *** (80)
Q7. Maintaining a culturally sensitive healthcare experience	5.8	0.89	6.51	0.69	−6.077 *** (80)
Q8. Speaking slowly	5.79	1.2	6.62	0.58	−6.141 *** (80)
Q9. Using plain nonmedical language	6.28	0.85	6.66	0.57	−3.592 ** (80)
Q10. Show or draw pictures	5	1.69	6.29	0.97	−6.754 *** (80)
Q11. Limit the amount of information provided and repeat it	5.64	1.22	6.46	0.81	−6.192 *** (80)
Q12.Use the teach-back or show me techniques	4.58	1.61	6.51 ()	0.74	−10.511 *** (80)
Q13.Create a shame-free environment	6.16	0.94	6.61	0.6	−4.698 *** (80)

*** *p* < 0.001, ** *p* = 0.01.

**Table 4 healthcare-11-00776-t004:** Self-Efficacy-12-Gr (SE-12-Gr) in pre- and post-assessments (*n* = 82).

Variable	Pre-Assessment		Post-Assessment		t
	Mean Score	SD	Mean Score	SD	
SE-12-Gr total	92.1	13.62	104.24	10.82	−11.13 ***
SE-1	77.4	11.22	78	8.2	ns
SE- 2	14.69	2.65	17.3	1.99	−10.41 ***

*** *p* < 0.001.

**Table 5 healthcare-11-00776-t005:** Self-Efficacy-12-Gr (SE-12-Gr) in pre-, post-, and follow-up assessments (*n* = 24).

Variable	Group	Rank Mean	Kruskal–Wallis
SE 12-GR SUM	Pre	26.42	8.990 *
	Post	43.92	
	Follow-up	39.17	
SE_1	Pre	32.33	6.313
	Post	31.92	
	Follow-up	45.25	
SE_2	Pre	24.08	14.313
	Post	46.15	
	Follow-up	39.27	

* *p* < 0.05.

**Table 6 healthcare-11-00776-t006:** Final adapted HLUPT for healthcare professionals working with older adults.

Final Adapted HLUPT Tool After Follow-Up	Content Modification
Tool 1. Raise awareness (path of improvement)	HLS-EU_Q16 added in annexes
	Personal experience from day center for older adults
	Planning seminars for older people visiting the day center
	Include points of healthy aging as part of the awareness presentation and address society, not only healthcare services
Tool 2. Communicate clearly (spoken communication)	Personal experience from day center for older adults
Tool 3. Teach-back (spoken communication)	Use of teach-back method for people with mild cognitive impairment—facilitate them with clinical appointment
Tool 4. Brown bag medicine reviews (spoken communication)	Personal experience from day center for older adults: facilitate older adults and carers to name their medicines as a memory game
Tool 5. Consider culture, customs, and beliefs, incl. points for language differences (spoken communication)	The role of rural communities and how they might influence older adults’ literacy
Tool 6. Assess, select, and create easy-to-understand materials (written communication)	No modification
Τοοl 7. Use health education material effectively (written communication)	No modification
Tool 8. Welcome patients: helpful attitude, signs, and more	Many healthcare services are not user-friendly; this tool may be useful for them
Tool 9. Encourage questions (self-management and empowerment)	Personal experience from day center and home care for older adults
Tool 10. Help patients remember how and when to take their medicine (self-management and empowerment)	Personal experience from day center for older adults
	Tips to help older people take their medication when their literacy level is low (e.g., draw images)
Tool 11. Get patient feedback (self-management and empowerment)	Added

## Data Availability

Data are unavailable due to privacy.

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
