# Peer review of "Cultural Adaptation of a Health Literacy Toolkit for Healthcare Professionals Working in the Primary Care Setting with Older Adults"

_healthcare, 2023, doi:10.3390/healthcare11050776_

Round 1

Reviewer 1 Report

The paper, "Cultural adaptation of a Health Literacy toolkit for health care professionals working in the primary care setting with older adults,"  focuses on the validation process of a complex tool, which may be of great interest to authors/researchers seeking knowledge in the area of translation and adaptation of other toolkits, and especially knowledge in the area in the area of Health Literacy, which is an area of real importance. The article has a rigorous and careful presentation. There is alignment between the title and the various sections of the document. The purpose of the study is clear. The introduction is straightforward and focused on the essentials. The methodological process of cultural adaptation is robust. The presentation of the method and results is clear. As a reviewer, I have one or another aspect that I would like to see clarified. The weakness of the article is its lack of conclusion.

As mentioned, in some respects, I would like additional information:

Comment 1

Line 112, when the authors refer "The testing methodology of HLUPT was based on the adaptation of the HLUPT for 112 Cardiology and Rheumatology [14]."

It's unclear to me. Do you mean that this method has already been used in adapting HLUPT to other clinical contexts? In this study, they are replicating a method already used (if that's the case, it seems positive to me). I ask that you make this information clearer.

Comment 2

Line 279 – the authors state that: "Almost half of the participants in the present study were not aware of the HL concept and identified only certain aspects of the HL concept (e.g., understanding health information, making health decisions, communication with physicians and physicians adherence) and not a thorough and coherent definition."

But in the results the authors mentioned that “More than two-thirds of the healthcare professionals (n=83, 64%) provided at baseline either the HL definitions or several dimensions, whereas the remaining gave vague definitions of health, skills, knowledge, clinical diagnosis, and communication skills of healthcare professionals. ”

This information seems contradictory to me. You can make this information clearer - " Almost half of the participants..."/ "More than two-thirds of the healthcare professionals (n=83, 64%) " ? 

Comment 3

It is recommended that the authors develop a conclusion where they mention the implications for the practice and for the investigation of this process of cultural adaptation of this tool.

Suggestions:

add the bibliographic sources, for example line 267 instead of [9], [18]–[21]; change to [9,18-21]

line 275 where is (23) change to [23]

line 56, 71, 73, 107 …269 – where is Heath literacy change to HL

Congratulation

Author Response

Thank you very much for your comments! Below there are point to point replies!

REVIEWER 1.

The paper, "Cultural adaptation of a Health Literacy toolkit for health care professionals working in the primary care setting with older adults,” focuses on the validation process of a complex tool, which may be of great interest to authors/researchers seeking knowledge in the area of translation and adaptation of other toolkits, and especially knowledge in the area in the area of Health Literacy, which is an area of real importance. The article has a rigorous and careful presentation. There is alignment between the title and the various sections of the document. The purpose of the study is clear. The introduction is straightforward and focused on the essentials. The methodological process of cultural adaptation is robust. The presentation of the method and results is clear. As a reviewer, I have one or another aspect that I would like to see clarified. The weakness of the article is its lack of conclusion.

As mentioned, in some respects, I would like additional information:

Comment 1

Line 112, when the authors refer "The testing methodology of HLUPT was based on the adaptation of the HLUPT for 112 Cardiology and Rheumatology [14]."

It's unclear to me. Do you mean that this method has already been used in adapting HLUPT to other clinical contexts? In this study, they are replicating a method already used (if that's the case, it seems positive to me). I ask that you make this information clearer.

Reply 1: Thank you for your comment, we have now rephrase this to: This pilot-test method has already been used in the adaptation of the HLUPT for Cardiology and Rheumatology

Comment 2

Line 279 – the authors state that: "Almost half of the participants in the present study were not aware of the HL concept and identified only certain aspects of the HL concept (e.g., understanding health information, making health decisions, communication with physicians and physicians adherence) and not a thorough and coherent definition."

But in the results the authors mentioned that “More than two-thirds of the healthcare professionals (n=83, 64%) provided at baseline either the HL definitions or several dimensions, whereas the remaining gave vague definitions of health, skills, knowledge, clinical diagnosis, and communication skills of healthcare professionals. ”

This information seems contradictory to me. You can make this information clearer - " Almost half of the participants..."/ "More than two-thirds of the healthcare professionals (n=83, 64%) " ? 

Reply 2: Thank you for your comment. We made this in accordance to the results

Two thirds of the participants in the present study were aware of the HL concept and identified aspects of the HL concept (e.g., understanding health information, making health decisions, communication with physicians and medical adherence)

Comment 3

It is recommended that the authors develop a conclusion where they mention the implications for the practice and for the investigation of this process of cultural adaptation of this tool.

Reply 3 : thank you, we have included conclusions

Comment 4. Suggestions: add the bibliographic sources, for example line 267 instead of [9], [18]–[21]; change to [9,18-21], line 275 where is (23) change to [23], line 56, 71, 73, 107 …269 – where is Heath literacy change to HL

Congratulation

            Reply 4. We have added the suggestions, but in the case of the reference brackets since it is from Mendeley they may change back to the original format.

Reviewer 2 Report

I think this is a fine study in an area deserving of increased attention (as we all age).  My only suggestion would be to add a bit in the discussion on reluctance (perhaps culturally based) of portions of the medical profession to adopt your recommendations and also add a short conclusion section to tie everything together.

Author Response

Thank you very much for your comments! Our replies follow:

REVIEWER 2

Comment 1. I think this is a fine study in an area deserving of increased attention (as we all age).  My only suggestion would be to add a bit in the discussion on reluctance (perhaps culturally based) of portions of the medical profession to adopt your recommendations

Reply 1: thank you very much, this has been added to the limitation of the study:

Limitations of this study are the small portion of medical professionals in the adaptation process and the small number of participants in the follow-up phase. During focus groups discussions with older healthcare users regarding their interaction with healthcare professionals, mostly focused on physicians instead of any other healthcare professionals

Comment 2. also add a short conclusion section to tie everything together.

Reply 2. We have added a conclusion section

Reviewer 3 Report

I have reviewed the manuscript entitled: "Cultural adaptation of a Health Literacy toolkit for health care professionals working in the primary care setting with older adults", in which the authors carry out an adaptation of a HL toolkit for healthcare professionals working with older adults. Although it is a work that can potentially be very relevant, it presents several issues throughout the sections that make up the manuscript. Below I mention some of them.

In the abstract (Lines 23-26), the description of the results is very general; the authors need to describe the main findings through statistical parameters adequately.

Lines 26-30. The way in which the information in the results is exposed does not allow the conclusions indicated by the authors .

It would be highly recommendable if the authors included in the introduction a section related to primary care and its importance in follow-up, continuity and first access to health services; as well as in the empowerment of patients in the management of the health-disease process.

In the Material and Methods section. Authors must follow the proper guidelines for reporting research in the medical field. For this reason, they must begin this section by defining appropriately: Type of study, population, informed consent and ethical aspects, geographical location, period in which the study was conducted, sample size, type of sampling, selection criteria (inclusion, exclusion and elimination (if applicable), hypothesis testing (if applicable), among others...

All tables contain grammatical errors (e.g. table 1 missing capital letters in subtitles, words together or misspelled, table 2, etc.)

Lines 84-88. The authors had to describe in more detail how they performed the reviews (e.g. databases, algorithms, filtering, etc.).

The authors should have included tables 2 and 3 in the results section. What is the reason for presenting results in a section that does not correspond to it?

The results section is difficult to follow, it is recommended that the authors create a flowchart where the most critical steps of the study are synthesized, the sample size and the methodological approach used (include statistical analysis). This information can replace Table 1.

Why do the authors put a table 4 and do not make a reference to it within the text? In addition, they place it just after citing table 5, this confuses the reader and gives a bad impression about the care they put authors when writing their work.

The authors should strengthen the discussion since it is concise and does not measure the main findings reported in the study, nor does it contrast them with the current information on the topic studied.

Authors need to incorporate a conclusions section.

Line 274. The authors need to note that these results apply only to the study population, since there is no external validity of the work (as the authors acknowledge in the limitations).

Author Response

Thank you very much for your comments. Below there are point to point comments:

REVIEWER 3

I have reviewed the manuscript entitled: "Cultural adaptation of a Health Literacy toolkit for health care professionals working in the primary care setting with older adults", in which the authors carry out an adaptation of a HL toolkit for healthcare professionals working with older adults. Although it is a work that can potentially be very relevant, it presents several issues throughout the sections that make up the manuscript. Below I mention some of them.

Comment 1. In the abstract (Lines 23-26), the description of the results is very general; the authors need to describe the main findings through statistical parameters adequately.

Reply 1; this was added in the case of the scale of Self-efficacy.

Comment 2. Lines 26-30. The way in which the information in the results is exposed does not allow the conclusions indicated by the authors .

Reply 2: this has been changed to be in accordance with the results section in the abstract

A culturally adapted HL toolkit was developed to support the needs of the healthcare professionals working with older adults, taking into consideration their feedback in all phases of the development.

Comment 3. It would be highly recommendable if the authors included in the introduction a section related to primary careand its importance in follow-up, continuity and first access to health services; as well as in the empowerment of patients in the management of the health-disease process.

Reply 3. Thank you this has been added in lines 48-59 (introduction)

Comment 4. In the Material and Methods section. Authors must follow the proper guidelines for reporting research in the medical field. For this reason, they must begin this section by defining appropriately: Type of study, population, informed consent and ethical aspects, geographical location, period in which the study was conducted, sample size, type of sampling, selection criteria (inclusion, exclusion and elimination (if applicable), hypothesis testing (if applicable), among others...

Reply 4. We have changed the structure of the method to be in accordance with the guidelines.

Comment 5. All tables contain grammatical errors (e.g. table 1 missing capital letters in subtitles, words together or misspelled, table 2, etc.)

Reply 5. We have corrected the Tables.

Comment 6. Lines 84-88. The authors had to describe in more detail how they performed the reviews (e.g. databases, algorithms, filtering, etc.).

Reply 6: thank you very much, this is already published to another journal and this is not the scope of the present manuscript. The publication is referenced in the paragraph. We added the four databases used and extended the results for the scope of this paper.

Comment 7. The authors should have included tables 2 and 3 in the results section. What is the reason for presenting results in a section that does not correspond to it?

Reply 7. We have transferred table 3 in the results section and now is table 2. Table 2 (now table 1 and 7) was separated in two tables: Table 1. HLUPT adaptation before piloting and Table 7. HLUPT final adaptation. In this way it is easiest to read and we do not include results in the methodology section.

Comment 8. The results section is difficult to follow, it is recommended that the authors create a flowchart where the most critical steps of the study are synthesized, the sample size and the methodological approach used (include statistical analysis). This information can replace Table 1.

Reply 8. We have replaced Table 1(phases of the study) with a flowchart. We did not complete the statistical analysis as this is described in relevant methodology section.

Comment 9. Why do the authors put a table 4 and do not make a reference to it within the text? In addition, they place it just after citing table 5, this confuses the reader and gives a bad impression about the care they put authors when writing their work.

Reply 9. Thank you we have corrected the numbering of the tables

Comment 10. The authors should strengthen the discussion since it is concise and does not measure the main findings reported in the study, nor does it contrast them with the current information on the topic studied.

Reply 10. We have strengthen the discussion with relevant literature on self-efficacy in communication and primary care access in Greece

Comment 11. Authors need to incorporate a conclusions section.

Reply 11. We have added the conclusion section

Comment 12. Line 274. The authors need to note that these results apply only to the study population, since there is no external validity of the work (as the authors acknowledge in the limitations).

Reply 12. This is now included in the limitation section.

Round 2

Reviewer 3 Report

I have read and reviewed version 2 of the manuscript: “Cultural adaptation of a Health Literacy toolkit for health care professionals working in the primary care setting with older adults”. In it, the authors have made several changes which allow readers to understand in greater depth the most important points of the study and how it was carried out.

One of the aspects in which the authors must work more, in future occasions, is in the strengthening of the discussion. The development of this section is fundamental during the interpretation and contextualization of the results; I invite you to review the following documents:

10.1186/s12874-018-0490-1

https://doi.org/10.1097/bsd.0000000000000687

10.5455/medarh.2018.72.306-307

PMID: 16733500

Author Response

Thank you very much for your review. We have advised all the proposed literature and we restructured discussion as follows: summary of key findings, novelty, similarities and controversies with previous research, strength and limitations, potential significance and future research